# Polymer cyclization for the emergence of hierarchical nanostructures

Chaojian Chen [1,2], Manjesh Kumar Singh [3], Katrin Wunderlich[1], Sean Harvey[1], Colette J. Whitfield [1], Zhixuan Zhou[1], Manfred Wagner[1], Katharina Landfester[1], Ingo Lieberwirth[1], George Fytas[1,4], Kurt Kremer [1], Debashish Mukherji[5], David Y. W. Ng [1✉] & Tanja Weil [1,2✉]

The creation of synthetic polymer nanoobjects with well-defined hierarchical structures is important for a wide range of applications such as nanomaterial synthesis, catalysis, and therapeutics. Inspired by the programmability and precise three-dimensional architectures of biomolecules, here we demonstrate the strategy of fabricating controlled hierarchical structures through self-assembly of folded synthetic polymers. Linear poly(2-hydroxyethyl methacrylate) of different lengths are folded into cyclic polymers and their self-assembly into hierarchical structures is elucidated by various experimental techniques and molecular dynamics simulations. Based on their structural similarity, macrocyclic brush polymers with amphiphilic block side chains are synthesized, which can self-assemble into wormlike and higher-ordered structures. Our work points out the vital role of polymer folding in macro-molecular self-assembly and establishes a versatile approach for constructing biomimetic hierarchical assemblies.

[1] Max Planck Institute for Polymer Research, Mainz, Germany. [2] Ulm University, Ulm, Germany. [3] Department of Mechanical Engineering, Indian Institute of Technology Kanpur, Kanpur, Uttar Pradesh, India. [4] Institute of Electronic Structure and Laser, Foundation for Research and Technology, Heraklion, Greece. [5] Quantum Matter Institute, University of British Columbia, Vancouver, Canada. ✉email: david.ng@mpip-mainz.mpg.de; weil@mpip-mainz.mpg.de

Precise three-dimensional architectures of biomacromolecules such as proteins and DNA have stimulated various new developments in macromolecular chemistry[1–4]. Intrigued by the programmability and specificity of intramolecular forces that enable folding of a giant molecular chain[5,6], synthetic chemists have taken several approaches to recreate, in part, notable features that contribute to these unique nanostructures. A simple biomimetic model typically involves a polymer chain consisting of chemical functions that recognize their interactive partner further along the chain, such that the main backbone can fold in a predictable way[7–10]. These functions funnel the free energy landscape of chain dynamics into, ideally, a single conformational region that would otherwise be subjected to randomness and kinetically trapped states. This led to the inception of single chain nanoparticles (SCNPs)[11], which focused on how polymers can be programmed to fold in a regular way akin to the chaperones of proteins[6]. The confinement characterized by these SCNPs has discovered newfound capabilities in catalysis, biomedicine, synthetic biology and when doped, offer interesting bulk material properties[12,13]. Despite these advancements, the larger perspective of how folding of a polymer chain can program higher ordered assemblies remains rare[14,15]. The concept of assembly driven by the regularity of macromolecules is a critical step for building sophisticated architectures that mimic the ability of proteins to form cellular nanostructures and compartments[16–19].

In this work, we focus on the emergence of structural complexity by folding of polymer chains forming unique cyclic structures capable of assembly into wormlike hierarchical structures. Polymers of 2-hydroxyethyl methacrylate (HEMA) are one of the simplest polymer scaffolds possessing a hydrophobic backbone and a hydrophilic side chain that promotes inter-chain interactions through van der Waals interactions and hydrogen bonds. Folding of the polymer is directed by a single copper-catalyzed azide–alkyne cycloaddition at the terminal ends[20]. The head-to-tail bite causes the polymer chain, in aqueous solvent, to adopt a self-propagating structure consisting of a hydrophobic core surrounded by the hydroxyl groups of HEMA. The physical properties and the complexity of the propagating structures can be subsequently controlled by growing a secondary block-copolymer on each HEMA side chain. Depending on the composition of the block copolymer, higher ordered assembly morphology between the unfolded and folded form can be achieved.

## Results

### Synthesis and cyclization of PHEMA homopolymers.

To demonstrate that synthetic polymers can be folded into specific forms for constructing higher ordered structures, polymers of HEMA (PHEMA) were folded into the cyclic topology, which is a well-established approach to fold polymers[21]. Three linear PHEMA samples (l-PHEMA$_n$-Br, where $n$ represents the number of repeating units) of different lengths were synthesized via atom transfer radical polymerization (ATRP) using propargyl 2-bromoisobutyrate as the initiator (Fig. 1a). Their respective average molecular weights were calculated from the $^1$H nuclear magnetic resonance (NMR) spectra as 1670 g mol$^{-1}$ ($n = 11$), 2120 g mol$^{-1}$ ($n = 15$), and 3040 g mol$^{-1}$ ($n = 22$), which fit well to the design (Supplementary Figs. 1–3 and Table 1). Gel permeation chromatography (GPC) results show relatively narrow molecular weight distributions of 1.36~1.39 (Fig. 1b). The apparent molecular weights determined by GPC are higher, which is due to the different hydrodynamic sizes of poly(methyl methacrylate) (PMMA) standards and PHEMA in N,N-dimethylformamide (DMF)[22]. Next, the bromine ends of these polymers were transformed to azide groups by reacting with sodium azide, which was confirmed by the appearance of the characteristic

peak of azide at 2121 cm$^{-1}$ in the Fourier-transform infrared (FTIR) spectra of the product (l-PHEMA$_n$-N$_3$) (Fig. 1c and Supplementary Fig. 4).

Subsequently, the heterobifunctional PHEMA with azide and alkyne ends was folded via Huisgen cycloaddition under high dilution conditions[20]. In a typical experiment, CuBr and 2,2′-bipyridyl (bpy) were added into a Schlenk tube loaded with DMF, which was degassed through two freeze–pump–thaw cycles. To avoid intermolecular reactions, the degassed DMF solution of l-PHEMA$_n$-N$_3$ was added into the catalyst solution at a slow speed of 0.16 mL h$^{-1}$ via a syringe pump. As shown in Fig. 1c, the asymmetric stretching peak of the azide group at 2121 cm$^{-1}$ vanished after this step. This indicates that the click reaction between azide and alkyne groups was successful, which was also proven by $^1$H NMR (Supplementary Figs. 5–7). We also proved the structure for all the products in each step by 2D $^1$H,$^1$H correlation spectroscopy and 2D $^1$H,$^1$H nuclear Overhauser effect spectroscopy methods (Supplementary Figs. 8–16).

The obtained polymers were further characterized by GPC (Fig. 1d and Supplementary Fig. 17) and $^1$H diffusion-ordered NMR spectroscopy (Supplementary Figs. 18–23). Both techniques demonstrate that the hydrodynamic volumes of all three samples decreased after the click reaction, which confirms that linear PHEMA polymers were intramolecularly folded into the cyclic topology. It should be mentioned that no shoulder peak in the higher molecular regions was observed in the GPC curves indicating no linear polymers left or oligomers formed. To investigate if a single f-PHEMA$_{15}$ (Supplementary Fig. 24) retains its well-defined cyclic structure in pure water, we performed an all-atom molecular dynamics simulation. In Fig. 1e we show the snapshot of a f-PHEMA$_{15}$ and the corresponding radius of gyration ($R_g$). To estimate the degree of sphericity of the cyclic polymers, we also calculated the lengths of the two major axes of the structure (Supplementary Fig. 25). The numbers are very similar over 20 ns, except for some small fluctuations within the time window of 12–14 ns. Therefore, the ring remains rather stable and symmetrically cyclic in pure water.

### Self-assembly of linear and cyclic PHEMA.

Linear PHEMA is generally regarded as a water-swellable polymer[23]. Using a dialysis method[24], we compared the self-assembly behaviors of linear and folded PHEMA. Briefly, the polymer was dissolved in methanol and then deionized water was added dropwise. The mixture solution was loaded into a dialysis membrane (MWCO = 1000 g mol$^{-1}$) and dialyzed against water. As shown in Fig. 2a and Supplementary Fig. 26, solutions of the linear polymers turned turbid after 1 h and some gel-like precipitation was observed 3 days later. Surprisingly, solutions of the folded polymers remain clear, a first indication that they can stabilize themselves in pure water.

Transmission electron microscopy (TEM) images reveal the formation of wormlike structures with a distribution of contour lengths in the range of 20–220 nm from the folded polymers (Fig. 2b and Supplementary Figs. 27–29). For linear polymers of the same molecular weights, only irregular aggregates were observed showing the significant role of folding in the self-assembly (Supplementary Figs. 27–29). We further confirmed the generation of stable assemblies from folded PHEMA by dynamic light scattering (DLS). Interestingly, both TEM and DLS results suggest f-PHEMA$_{15}$ with a medium ring size formed the smallest assemblies. The rate $\Gamma(q)$ obtained from the representation of the relaxation function $C(q,t)$ (Supplementary Figs. 30–32) is purely diffusive ($\Gamma(q) = Dq^2$). Hence the $q$-independent translational diffusion $D$ is free of internal assembly dynamics allowing the determination of the hydrodynamic radius $R_h$ for assemblies of

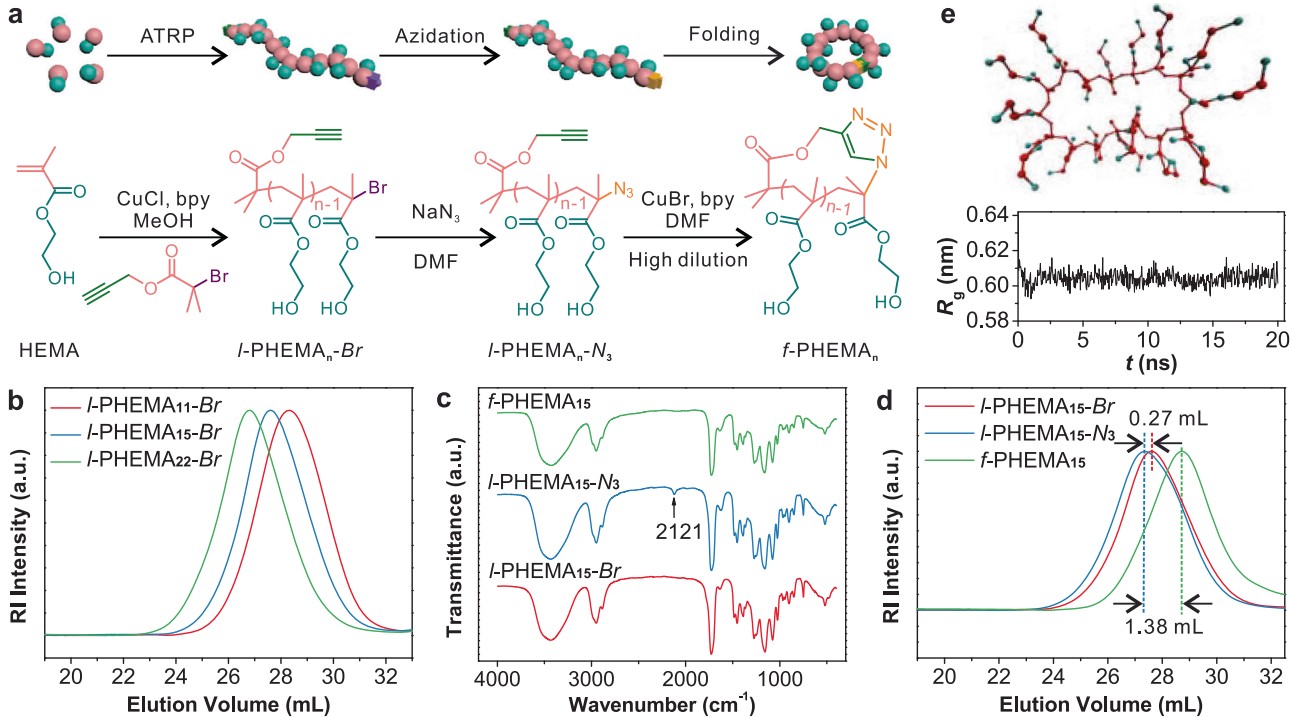

**Fig. 1 Synthesis and folding of PHEMA. a** Scheme of the synthesis of *l*-PHEMA$_n$-*Br* and its azidation and folding into the cyclic topology. **b** GPC curves (eluent: DMF; standard: PMMA) of *l*-PHEMA$_n$-*Br* with different repeating units. **c** FTIR spectra, and **d** GPC curves of *l*-PHEMA$_{15}$-*Br*, *l*-PHEMA$_{15}$-*N$_3$*, and *f*-PHEMA$_{15}$. **e** The top and bottom panels show a simulation snapshot and the time evolution of $R_g$ of a single *f*-PHEMA$_{15}$ in pure water, respectively.

### Table 1 Characteristics of linear and folded PHEMA samples.

| Entry | $M_{n, \text{design}}$ (g mol$^{-1}$) | $M_{n, \text{NMR}}$[a] (g mol$^{-1}$) | $M_{n, \text{GPC}}$[b] (g mol$^{-1}$) | $M_w$[b] (g mol$^{-1}$) | $V_p$[b] (mL) | $\text{Đ}$[b] ($M_w/M_n$) |
|---|---|---|---|---|---|---|
| *l*-PHEMA$_{11}$-*Br* | 1500 | 1670 | 6400 | 8800 | 28.30 | 1.39 |
| *l*-PHEMA$_{11}$-*N$_3$* | / | / | 5700 | 8800 | 28.21 | 1.54 |
| *f*-PHEMA$_{11}$ | / | / | 4500 | 6000 | 29.16 | 1.33 |
| *l*-PHEMA$_{15}$-*Br* | 2150 | 2120 | 8300 | 11300 | 27.62 | 1.36 |
| *l*-PHEMA$_{15}$-*N$_3$* | / | / | 8600 | 12800 | 27.35 | 1.49 |
| *f*-PHEMA$_{15}$ | / | / | 5000 | 7400 | 28.73 | 1.47 |
| *l*-PHEMA$_{22}$-*Br* | 3450 | 3040 | 11400 | 15900 | 26.80 | 1.39 |
| *l*-PHEMA$_{22}$-*N$_3$* | / | / | 10900 | 15900 | 26.77 | 1.45 |
| *f*-PHEMA$_{22}$ | / | / | 6800 | 7900 | 28.09 | 1.17 |

[a]Determined by $^1$H NMR. [b]Determined by GPC using DMF as eluent and PMMA as standards.

*f*-PHEMA$_{11}$, *f*-PHEMA$_{15}$, and *f*-PHEMA$_{22}$ amounting to 54 ± 2, 36 ± 2, and 58 ± 3 nm, respectively (Fig. 2c and Supplementary Figs. 30–32). The large size of the *f*-PHEMA$_{11}$ and *f*-PHEMA$_{22}$ assemblies was confirmed by their measurable $R_g$ in the Ornstein–Zernike, $I(q)^{-1}$ vs $q^2$ (lower inset to Fig. 2c): $I(q)$ of the smaller *f*-PHEMA$_{15}$ case is virtually $q$ independent[25]. The calculated $R_g =$ 50 nm (59 nm) for the *f*-PHEMA$_{11}$ (*f*-PHEMA$_{22}$) assemblies are very similar to the values of $R_h$ suggesting compact structures much larger than the single rings (Fig. 1e). For linear PHEMA, DLS experiment was not possible due to the presence of large aggregates even at low number densities as indicated by the TEM data (Supplementary Figs. 27–29).

The assembly behaviors of cyclic PHEMA can be ascribed to the rearrangement of the atomic distribution by polymer folding. Nile Red loading experiments indicated the formation of hydrophobic microenvironments during the assembly (Supplementary Fig. 33). The measured critical aggregation concentrations of *f*-PHEMA$_n$ (~0.01 mg mL$^{-1}$) are lower than that of many amphiphilic polymers (0.02~0.2 mg mL$^{-1}$)[24,26], showing the strong assembly trend of folded polymers. Furthermore, we

performed NMR measurements for *f*-PHEMA$_n$ in solvent mixtures of methanol-d$_4$ (MeOD) and deuterium oxide (D$_2$O) with gradually tuned compositions (Fig. 2d and Supplementary Fig. 34). When the amount of D$_2$O increased, the integrated intensity for the signals of the methyl groups in the backbone decreased, which is a signature of reduced backbone mobility due to the onset of self-assembly, and their positions did not change. However, the peaks from the methylene groups (3.77 and 4.03 ppm) in hydroxyethyl side chains shifted. This result indicates that the backbones of *f*-PHEMA$_n$ were packed in the core and the side chains on the surface enabling them to freely interact with the polar solvent.

To further elucidate the self-assembly, we performed molecular dynamics simulations of an implicit solvent generic model. In this model, the quantities are expressed in the units of energy $\varepsilon$, length $\sigma$ and time $\tau$ (see the Supplementary information, pp. 12, 13). The choice of the generic model parameters of a ring polymer is inspired by the structural stability and the solubilities of the individual residues in pure water, as observed for a *f*-PHEMA$_{15}$ (Fig. 1e)[27,28]. For this purpose, we have investigated two ring sizes ($n = 9$ and 15) at the same mass density. Due to the hydrophobic attractions between the backbone rings, these molecules self-assemble into the wormlike micellar structures, see the simulation snapshots in Fig. 2e, f. In our case, while we abstain from discussing the details of aggregation kinetics, we note that the sizes of the largest aggregates stabilize after a time $t \sim 10^4 \tau$ in both cases. For example, the length of the largest aggregate for $n = 9$ is about $L \sim 33.5\sigma$, while $L \sim 14.2\sigma$ for $n = 15$. The size of the aggregates decreases going from $n = 9$ to $n = 15$, as also evident from the experimental data in Fig. 2b, c. This behavior is consistent with the fact that the simulations (as the experiments) are performed at a constant mass density for different $n$ and thus will automatically infer that the number density $\rho$ of the molecules decreases with increasing $n$. Here, a self-assembled

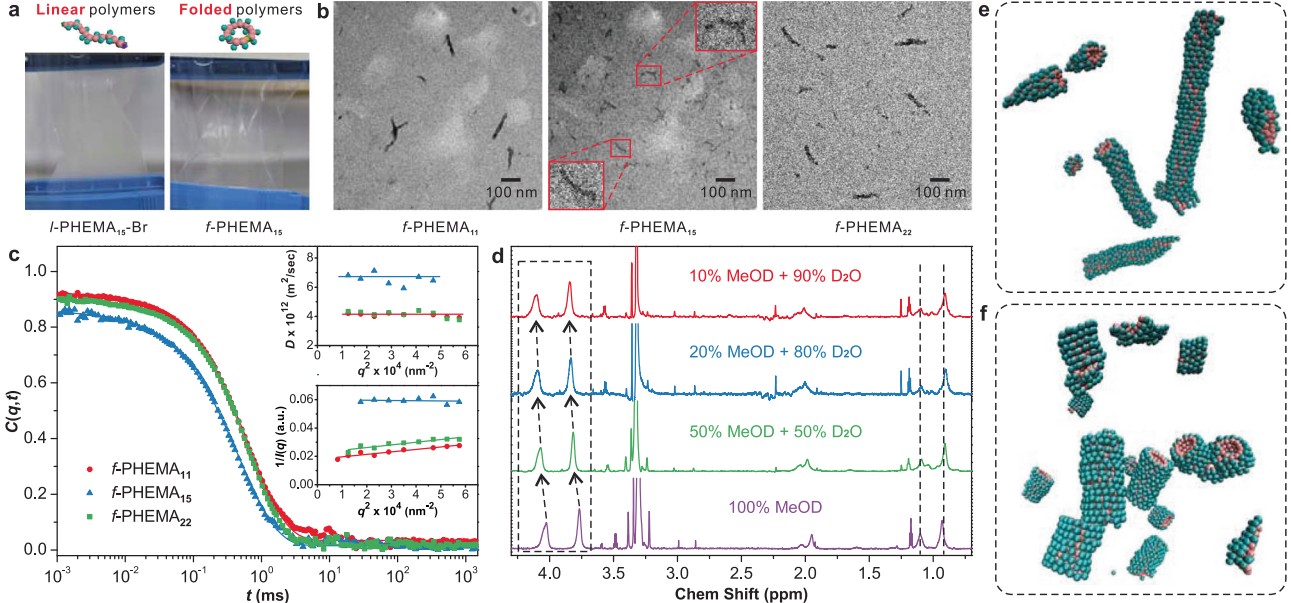

**Fig. 2 Self-assembly of linear and folded PHEMA. a** Solutions of *l*-PHEMA$_{15}$-*Br* and *f*-PHEMA$_{15}$ after dialysis against deionized water for 1 h. **b** TEM images showing the assemblies of *f*-PHEMA$_{11}$, *f*-PHEMA$_{15}$, and *f*-PHEMA$_{22}$. **c** Relaxation functions $C(q,t)$ for the translation motion of *f*-PHEMA$_{11}$, *f*-PHEMA$_{15}$, and *f*-PHEMA$_{22}$ in water at 1 mg mL$^{-1}$ and 293 K at a scattering angle of 90° corresponding to scattering wave vector $q = 0.0187$ nm$^{-1}$. Upper inset: The translation diffusion coefficient, $D$ as a function of $q^2$ with the solid line indicating a virtually $q$-independent $D$. Lower inset: $1/I(q)$ as a function of $q^2$. **d** $^1$H NMR spectra (850 MHz, 298.3 K) of *f*-PHEMA$_{15}$ in mixtures of MeOD and D$_2$O with gradually tuned volume ratios. The chemical shifts were calibrated using tetramethylsilane as an internal standard. **e, f** Molecular dynamic simulation snapshots showing the self-assembly of a model simulation replica of *f*-PHEMA$_n$ with two different ring sizes, i.e., $n = 9$ (**e**) and $n = 15$ (**f**).

structure is dictated by the competition between the entropy penalty of forming an aggregate of a particular size from the homogeneous mixture and the surface energy reduction[29], which is intimately linked to $\rho$. In the experiments, the aggregate size again increases for $n = 22$. This can be attributed to the peripheral ring fluffiness at $n = 22$. Therefore, it is expected that the assemblies form rather mushy objects with relatively collapsed hydrophobic cores and not any well-defined wormlike micelles.

From $L$ and the radius ($r$) for the simulated structures in Fig. 2e, f, we have also estimated $R_g$ and $R_h$ using the approximate expressions, i.e., $R_g = \left(\frac{L^2}{12} + \frac{r^2}{2}\right)^{1/2}$ and $R_h = \frac{L}{2s - 0.19 - \frac{8.24}{s} + \frac{12}{s^2}}$, with $s = \ln(L/r)$ [30]. Following these estimates, we find $R_g/R_h \sim 1.28$ for $n = 9$ and 0.85 for $n = 15$. Here, it is known that $R_g/R_h \sim 1.5$ for a coil and $(3/5)^{1/2}$ for a spherical globule[31]. In our simulations, systems with both $n$ values fall within these two limits, while $n = 15$ shows $R_g/R_h$ behavior closer to $(3/5)^{1/2}$ that is somewhat expected given that the structures are rather short and thick.

It should also be noted that the structures observed in our simulations reveal the microscopic picture of the single aggregates. Moreover, from the assemblies in Fig. 2b, they typically have a diameter of 10~20 nm. Therefore, considering that the estimated diameter of a ring is about 1 nm (Supplementary Fig. 25), each structure in Fig. 2b is expected to consist of 10~20 wormlike micelles stacked sideways to form a bundle. The microscopic description of such stacking in good solvent is rather well established in the case of entropically stabilized elongated objects[32].

**Hierarchical self-assembly of cyclic brush polymers**. The above results clearly show that even homopolymers with a very simple structure can assemble into higher ordered nanostructures after molecular folding. Therefore, we believe, like polypeptides and nucleic acids, common synthetic polymers can also be programmed into controlled nano-objects with multiple levels of

defined architectures through folding and assembly. As a proof-of-concept, we designed macrocyclic brush polymers with amphiphilic polystyrene-*block*-poly(acrylic acid) (PS-*b*-PAA) side chains. PS and PAA were selected because they are broadly representative of hydrophobic and hydrophilic polymers, and PS-*b*-PAA is one of the most intensively studied systems in the field of macromolecular self-assembly[26].

The cyclic macroinitiator *f*-P(HEMA-*Br*)$_{22}$ was synthesized by attaching ATRP initiators to hydroxyl groups of *f*-PHEMA$_{22}$ (Fig. 3a). Hydrophobic PS and poly(*tert*-butyl acrylate) (P*t*BA) were consecutively grafted from the cyclic macroinitiator. The P*t*BA block was then hydrolyzed by trifluoroacetic acid into PAA, generating cyclic brush polymers with amphiphilic side chains (*f*-P(HEMA-*g*-PS$_x$-*b*-PAA$_y$)$_{22}$, where $x$ and $y$ represent the numbers of repeating units for PS and PAA, respectively). These cyclic brush polymers can be regarded as the simplest folded form of the corresponding linear brush polymers. As shown in Fig. 3b, c, we synthesized four cyclic brush polymers (**CB-1** to **CB-4**). For comparison, a block copolymer (**BC**) and a linear brush polymer (**LB**) with comparable compositions were also prepared. All products in each step have been systematically characterized by NMR and FTIR spectroscopies as well as GPC (Supplementary Figs. 35–49 and Supplementary Tables 1–3). Analysis of cyclic brush polymers by GPC shows narrow size distributions with Đ in the range of 1.17–1.48, indicating that they can be used as uniform building blocks.

Using a similar dialysis method, we investigated the self-assembly behavior of these amphiphilic cyclic brush polymers (**CB-1** to **CB-4**) in water and compared them with **BC** and **LB**. The polymers were first dissolved in DMF, which is a good solvent for both PS and PAA. After adding the same amount of deionized water dropwise, the mixture solutions were dialyzed against water for 3 days and then tuned to desired concentrations. As shown in Fig. 3d and Supplementary Fig. 50, block copolymer **BC** self-assembled into vesicles and large compound vesicles at

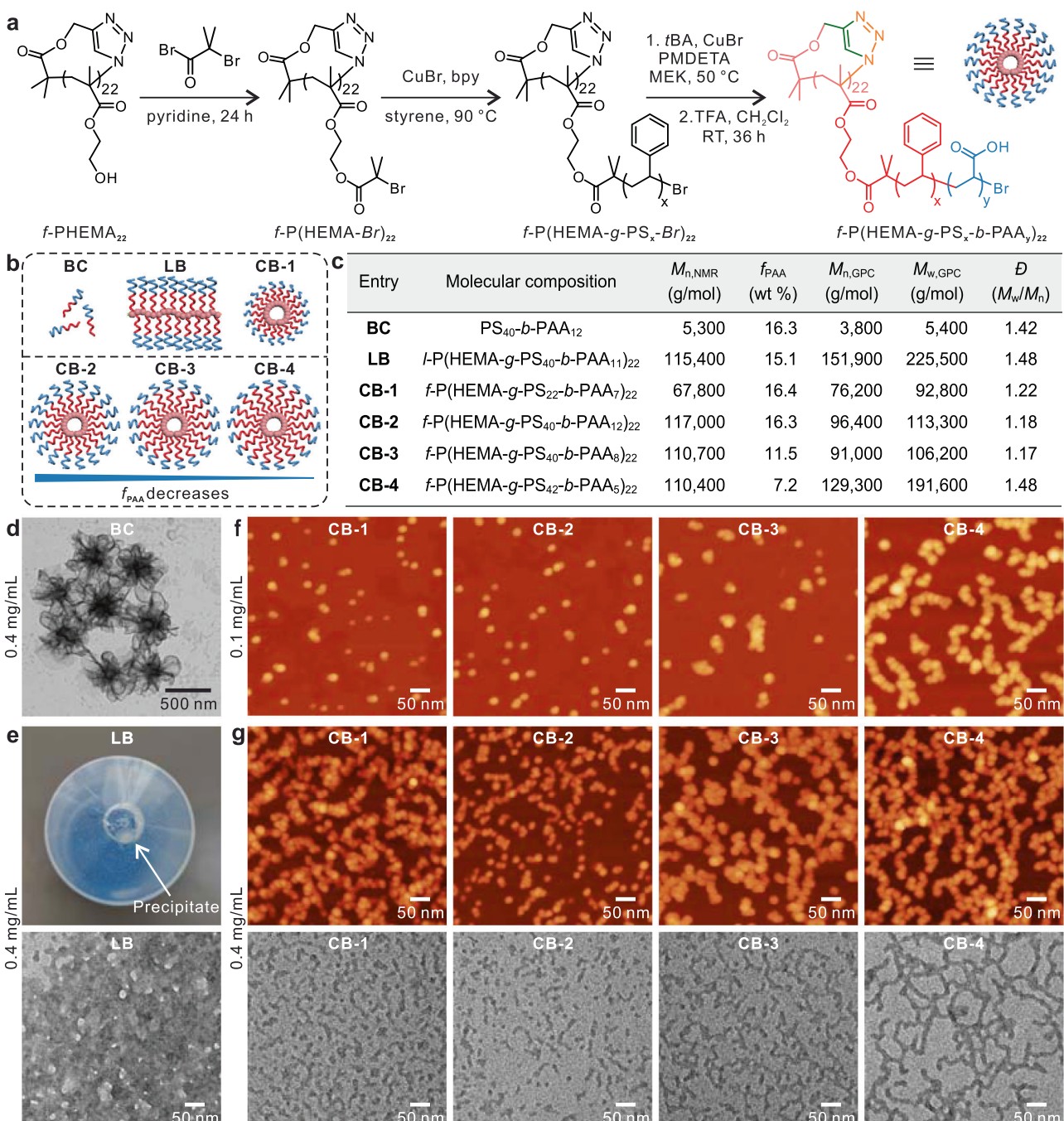

**Fig. 3 Hierarchical self-assembly of cyclic brush polymers. a** Scheme for the synthesis of cyclic brush polymers $f$-P(HEMA-$g$-PS$_x$-$b$-PAA$_y$)$_n$. **b** Schematic illustration, and **c** molecular parameters of block copolymer **BC**, linear brush polymer **LB**, and four cyclic brush polymers (**CB-1** to **CB-4**). **d** TEM image showing the self-assembly of **BC** at 0.4 mg mL$^{-1}$ in water. **e** Optical (top) and TEM (bottom) images showing the aggregation of **LB** in water. **f** AFM images showing the self-assembly of **CB-1** to **CB-4** at 0.1 mg mL$^{-1}$ in water. **g** AFM (top) and TEM (bottom) images showing the self-assembly of cyclic brush polymers into 1D wormlike assemblies and hierarchical structures at 0.4 mg mL$^{-1}$ in water.

different concentrations due to its high weight fraction of hydrophobic PS. In comparison, the **LB** counterpart to the polymerized form of **BC** could not form ordered structures and precipitation was observed at 0.4 mg mL$^{-1}$ (Fig. 3e and Supplementary Figs. 51–52).

Interestingly, cyclic brush polymers with similar compositions but a compact topology demonstrated drastically different self-assembly behaviors. Although the nature of the block copolymer (**BC**) and its linear polymerized form (**LB**) exhibit low solubility even in DMF featuring large aggregates upon addition of water,

the cyclization is able to alleviate these effects and create stable assemblies. We found that the polymer concentration and the weight fraction of PAA ($f_{PAA}$) are two important factors on the assembly. Discrete structures were observed by atomic force microscopy (AFM) and TEM for all samples when the concentrations were 0.1 mg mL$^{-1}$ (Fig. 3f). The smallest particles in the AFM images show the same height of about 5 nm (Supplementary Figs. 53–55), which corresponds with the size of single cyclic brushes. Other bigger nano-objects can be ascribed to dimers, trimers, or oligomers of cyclic brush polymers.

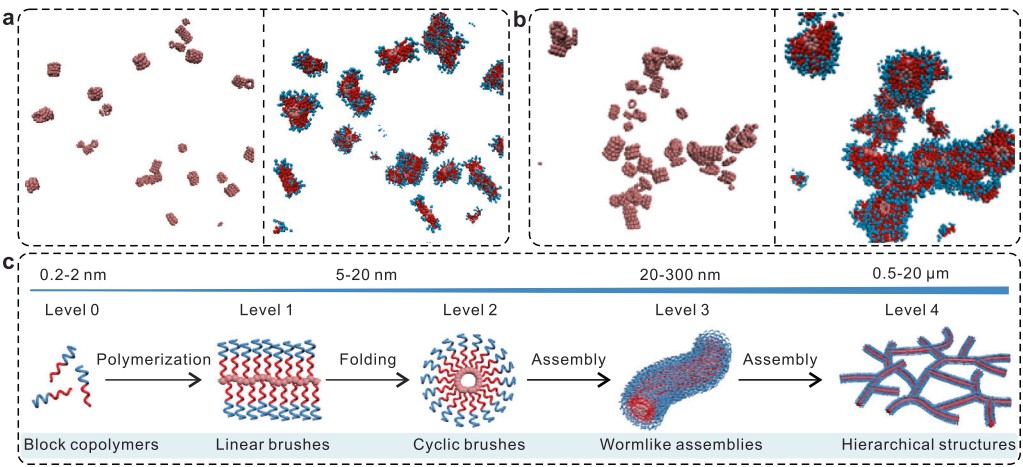

**Fig. 4 Emergence of structural complexity by self-assembly of cyclic brush polymers. a, b** Molecular simulation results showing the self-assembly of cyclic brush polymers for the two different concentrations, $\rho = 0.0002\sigma^{-3}$ (**a**) and $0.0009\sigma^{-3}$ (**b**). The left panels show only the hydrophobic backbones and the right panels illustrate the full molecules. **c** Formation of hierarchical structures by synthetic polymers via biomimetic folding and self-assembly.

Importantly, the boundary between cyclic brush polymers can be visualized by AFM, which clearly shows the assembly of cyclic brush polymers in a layer-by-layer manner (Fig. 3f and Supplementary Figs. 53–56). When the concentrations increased to 0.4 mg mL$^{-1}$, wormlike assemblies and hierarchical structures were obtained (Fig. 3g and Supplementary Figs. 57–60). To further prove the formation of wormlike nanostructures in solution, we performed cryo-TEM measurement for **CB-4**. The images in Supplementary Fig. 61 clearly confirm that branched wormlike assemblies were generated in solution. In addition, these wormlike hierarchical structures demonstrated good stability under various harsh conditions such as staining and washing, sonication for 1 h, or storage at room temperature for 2 months (Supplementary Figs. 62–64). Unlike simple diblock copolymers that form wormlike micelles only in a narrow window of hydrophobic–hydrophilic ratios, folded polymers with a broad range of compositions can assemble into wormlike structures. More importantly, these wormlike structures are generated via stepwise folding and modular assembly, allowing the modulation of the internal structure and overall dimension in each step. For cyclic brush polymers with similar side chain lengths (**CB-2** to **CB-4**), the assemblies became longer with the decrease of $f_{PAA}$. By customizing the block constituents, proportion and length, the architectural outcome can be changed.

To investigate these structures, we have also performed a set of generic simulations of a replica of the cyclic brush polymers for five different concentrations $\rho$. The data for two concentrations is shown in Fig. 4a, b and the complete data is shown in Supplementary Fig. 65 and Supplementary Movies 1–10. Here, the presence of side chains greatly hinders the self-assembly. For $\rho = 0.000\sigma^{-3}$, discrete assemblies are observed attaining a maximum size of ~7.0$\sigma$ (Fig. 4a) that is about 5 times smaller than the model $f$-PHEMA$_9$ with $n = 9$ without the long amphiphilic side chains (Fig. 2e). Furthermore, the number of backbone molecules stacked together in an aggregate decreases with increasing $c$ and also the side chains start to interdigitate attaining almost a network-like assembly for $\rho \geq 0.0005\sigma^{-3}$, see Fig. 4a, b and Supplementary Fig. 65. These observations are consistent with the AFM and SEM measurements. Similar to the $f$-PHEMA system, the stacking is still dictated by the hydrophobic attraction between the backbone rings, while the interdigitation is because of the polystyrene–polystyrene hydrophobic contacts between the side chains of the neighboring microstructures.

## Discussion

We have therefore established a biomimetic strategy for the construction of hierarchical nanostructures via self-assembly of folded polymers. By rearranging atomic distribution of polymers to form specific preliminary structures, this modular approach is particularly powerful for preparing wormlike assemblies from synthetic polymers including common homopolymers and block copolymers. Various parameters including the polymer composition, molecular weight, and primary structure can be used to manipulate the overall structure of the assemblies. The self-assembly pathway for the hierarchical structures is a versatile and stage-distinct platform to mimic the increasing complexity in the folding of polypeptides into 3D precise protein structures (Fig. 4c). The combination of polymer folding and self-assembly can therefore serve as an effective avenue for fabricating next-generation hierarchical structures with significantly increased complexity which cannot be achieved by traditional methods.

## Data availability

All of the data that support the findings of this study are available within the article and its Supplementary Information files.

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

## Acknowledgements

The authors acknowledge financial support from Deutsche Forschungsgemeinschaft (DFG, German Research Foundation) – Project number 213555243 – SFB 1066 (A6). C.C. is grateful for a doctoral fellowship from Promotionskolleg Pharmaceutical Bio-technology of Ulm University funded by the state of Baden-Württemberg. D.M. thanks the Canada First Research Excellence Fund (CFREF) for financial support. M.K.S. thanks generous allocation of computational time at the IIT Kanpur computer facility where most generic simulations were performed. D.M. thanks the ARC Sockeye super-computing facility where the all-atom simulations were performed. G.F. acknowledges the financial support by ERC AdG SmartPhon (Grant No. 694977).

## Author contributions

T.W. acquired funding for the project. T.W. and D.Y.W.N. supervised the project and corrected the manuscript. C.C. initiated the idea, performed the experiments, and wrote the manuscript. M.K.S., K.K., and D.M. performed the molecular simulations. K.W. and G.F. interpreted the light scattering results. S.H. and C.J.W. performed the AFM. Z.Z. assisted with the self-assembly of **CB-4**. I.L. and K.L. conducted the cryo-TEM analysis. M.W. provided the NMR expertise.

## Funding

## Competing interests

The authors declare no competing interests.
