## [Peer Review File. · Nature Communications]

REVIEWER COMMENTS

Reviewer #1 (Remarks to the Author):

The work described in this paper is inspired by the ability of some biomolecules to form well-defined hierarchical structures, and the authors take steps to achieve that goal with synthetic polymers. The following features are important to their design:

i) a modest degree of polymerization (DP), between 10 and 25; ii) a hydrophobic backbone with hydrophilic pendant groups; iii) a polymer that is borderline insoluble in water, i.e., swellable by water but not soluble; iv) introduction of end-group functionality and ring closure to a cyclic form (which the authors call "folding"). For this purpose, they synthesize poly(hydroxyethyl methacrylate) (PHEMA) by ATRP with azide and alkyne end groups with number average DP_n values of 11, 15, 22, (denoted I-PHEMA) and use a high dilution method to obtain the cyclic polymers (denoted f-PHEMA). The dispersities are only modestly narrow ($\text{Đ} = 1.35 - 1.39$) and the GPC curves in Figure 1B show a considerable overlap in composition.

The authors then compare what happens with the linear polymer and the folded polymer when methanol solutions are dialyzed against water. The main difference is that the linear polymer precipitates whereas f-PHEMA forms colloidally stable aggregates. The authors characterized these aggregates by TEM in the dry state on a grid, as well as multiangle dynamic light scattering and X-ray scattering to obtain the hydrodynamic radius R_h and the radius of gyration R_g in solution. These very careful experiments are complemented by coarse-grained molecular dynamics simulations of the conformation and the assembly of a cyclic structure. The submission is accompanied by a number of pretty videos showing interaction of these model structures.

After analysis of this first set of experiments, the authors attach an ATRP initiator to all of the -OH groups of f-PHEMA and grow a series of polystyrene-block-poly(acrylic acid) (PS-b-PAA) pendants from the ring. The assembly of these graft copolymers was compared to that of the corresponding I-PHEMA graft copolymer. Again, the syntheses seem to have been very carefully executed and there is a high level of polymer characterization.

Critical comments:

1. The TEM images of the assemblies formed by f-PHEMA in water show elongated structures that are polydisperse in length and variable in width. It would be helpful to the reader if the authors could provide histograms of the length and width distributions. The simulations (Figure 2 E,F) also show a broad dispersity of aggregate sizes and shapes.

2. It is curious that these structures appear to be characterized by a diffusion coefficient D that is invariant with q^2 (Fig 2C), a behavior expected for spheres. Do the authors have multi-angle static light scattering data for these systems?

3. The structures formed by the DP_n = 15 polymer was much smaller than those formed by the DP_n = 11 and 22 polymers. I find this surprising and the authors do not comment on the reproducibility of this result. Some authors (e.g., Chi Wu) use the ratio of R_g/R_h to obtain structural information about objects in solution. Here the authors make no comment about the finding that $R_g/R_h \sim 1$. Another feature of the data that the authors do not comment about is the reduction of the integrated intensity in the ¹H NMR spectra of the backbone methyl groups of f-PHEMA₁₅ in mixtures of D₂O and MeOD as the D₂O content is increased (Figure 2B). This may be a signature of reduced backbone mobility, perhaps as a consequence of the onset of association.

3. The self-assembly results for the series of f-P(HEMA-g-PS-b-PAA) cyclic graft copolymers shown in Figure 3G is very interesting. Significant differences were found as a function of PS and PAA composition, but all are different from that of the linear analogue.

4. All of the structures formed by aggregation in water are complex and hard to describe. This does not diminish the value of these experiments, but one has to be cautious about how one describes this self-assembly process.

For example, for f-PHEMA in water, given the relatively broad dispersity, one should probably

conclude that the association of these cyclic forms accommodates a broad distribution of contour lengths.

Then there are examples where word choice is out of place. For example, in the title, this is a strategy for the emergence of hierarchical nanostructures, but it is not a "general" strategy. The statement on p 2 "folding of polymer chains forming specific secondary structures capable of controlled assembly into anisotropic hierarchical structures" implies a much greater level of control than the authors have achieved. On p 3, the authors write "The physical properties and the complexity of the propagating structures can be subsequently customized (my italics) by growing a secondary block-copolymer on each HEMA side chain." It is clear that the secondary block copolymer changes aspects of the self-assembly, but I do not see a design rule that enables the authors to choose a block copolymer composition to reach a predetermined association structure. On the same page, they add "Depending on the composition of the block-copolymer, higher ordered assembly morphology between the unfolded and folded form can be programmed (my italics). Then on page 12 they write "By customizing the block constituents, proportion and length, the architectural outcome can be tuned (my italics). These are unnecessary exaggerations of what they can achieve.

In summary, I like this work. It is an interesting new idea. The execution required careful polymer synthesis, as well as careful and thorough characterization of the assemblies formed. The molecular dynamics simulations were a helpful complement to the experimental work. It represents an important and useful step in addressing an important challenge in the generation of hierarchical structures by synthetic polymers. I recommend acceptance after minor revisions, and I hope that the authors can be more precise in describing what they actually accomplished.

Reviewer #2 (Remarks to the Author):

In this work, Chen et al. presents an interesting approach to prepare (hierarchical) nanostructures via the self-assembly of cyclic polymers/cyclic polymer brushes. The authors started off by showing that a series of short chain cyclic PHEMA polymers (f-PHEMA) can be subjected to a (conventional) dialysis self-assembly process to yield wormlike nanostructures. To highlight the importance of the cyclic polymer topology, the authors showed that linear PHEMA counterparts (l-PHEMA) with similar block lengths cannot be used to access wormlike morphologies. The self-assembly process was scrutinized using various experimental techniques and further elucidated using molecular dynamics simulations. Here, the self-assembly process is presumably driven by phase separation of the hydrophobic methacrylate backbone and the hydrophilic hydroxyl side chains.

In the latter section of the manuscript, the authors expanded on their concept to prepare more structurally complex nanostructures or so-called "hierarchical nanostructures". This was achieved by, first, converting the cyclic PHEMA polymers (f-PHEMA) into cyclic ATRP macroinitiators, before polymerizing them into cyclic brush polymers (f-P(HEMA-g-PS-b-PAA)). Due to the amphiphilicity of the resulting cyclic brush polymers (f-P(HEMA-g-PS-b-PAA)), they exhibit a different self-assembly behaviour than with cyclic PHEMA polymers (f-PHEMA). Subjection to the same self-assembly procedure (i.e., dialysis from DMF to water) led to the formation of branched wormlike nanostructures. Negative control polymers with different polymer architectures (e.g., linear/linear brushes) were shown to form other non-wormlike morphologies under the same self-assembly conditions. The reported structures here are highly novel, but some further analysis is required to prove their existence.

Below are my comments:

Page 7, line 5 – can the authors comment on why the middle ring size gave the smallest R_h values? Looking at Table 1, could this be an effect of the differences in polymer dispersity? f-PHEMA15 has a much higher \bar{M}_w than the other two folded polymers.

Page 7, line 14 – the authors argue that DLS is not feasible as a characterization technique because the aggregates formed by l-PHEMA are too large. Their TEM data in Figure S27 (~200 nm particles) and Figure S28 (<1 μm), however, suggests otherwise?

Page 11, self-assembly process/data – I am a bit confused as to why the self-assembly was

carried out in this manner. Typically, when investigating the effect of polymer concentration on polymer self-assembly, one would start with molecularly dissolved polymer solutions at different concentrations and induce self-assembly (e.g., by dialysis against water such as in this work) to generate different polymer morphologies. Here, however, the authors performed self-assembly using a fixed initial polymer concentration of 1.6 mg/mL (according to SI) and added 50 vol% of water (nonsolvent) to the polymer/DMF solution. They then dialyzed this mixture against water and finally adjusted the samples to two concentrations, either to 0.1 mg/mL or 0.4 mg/mL where they observed different apparent morphologies. In some ways, what the authors have done here is they simply started off by generating some form of nanostructure via dialysis at relatively high concentrations (~ 0.8 mg/mL according to the SI) and then subsequently diluting the nanostructures down to 0.1 mg/mL or 0.4 mg/mL. The data presented in Fig. 3 therefore does not adequately highlight the influence of polymer concentration on self-assembly, but rather points towards the fact that the nanostructures formed post-dialysis at ~ 0.8 mg/mL are non-equilibrium structures that dissociate into smaller subunits (apparent as spheres or “nanoobjects” as the authors refer to) at high dilution factors.

Most importantly, I am currently not sure if the branched wormlike nanostructures are real since the authors have only presented microscopy (TEM/AFM) data at relatively high concentrations where the drying effect in TEM/AFM sample preparation is known to cause particles to “artificially” agglomerate with one another. If possible, the authors should at least present some microscopy data of the particles in their native solution state (e.g., cryo-EM or confocal/fluorescence microscopy; the latter should be easy to conduct since the authors showed in Fig. S33 the possibility of loading a fluorescent dye into their particles). Photographs of the sample solutions would also be very helpful here as highly branched wormlike structures are known to appear as insoluble precipitates in solution that are visible to the naked eye. Also, I am curious as to why no DLS data was provided for the nanostructures in Fig. 3. I find this very surprising since DLS was used to thoroughly characterize the wormlike nanostructures formed by f-PHEMA (see Fig. 2C and Fig. S32).

Regarding the self-assembly of the cyclic polymer brushes – Can the authors comment on the formation mechanism behind the branched wormlike nanostructures? I struggle to understand how individual polymer subunits (i.e., nanoobjects in Fig. 3F/G) would preferentially stack in an anisotropic fashion form cylinders/worms or even branch out into fractal structures. These polymer subunits are supposedly decorated with a PAA brush corona. Wouldn't the particles prefer to repel one another due to electrostatic repulsion in addition to the brush conformation?

At the present stage, I find the first half of the manuscript to be well-written and substantiated by convincing data. However, the second half of the manuscript (in particular, the section on the self-assembly cyclic polymer brushes) needs more data. I strongly suggest the authors perform (i) additional characterization on their so-called “wormlike assemblies” and “hierarchical structures” to refute the possibility that these morphologies are imaging artifacts, and (ii) elaborate on their formation mechanism, if they are indeed real.

Point by Point Response

Reviewer #1 (Remarks to the Author):

The work described in this paper is inspired by the ability of some biomolecules to form well-defined hierarchical structures, and the authors take steps to achieve that goal with synthetic polymers. The following features are important to their design: i) a modest degree of polymerization (DP), between 10 and 25; ii) a hydrophobic backbone with hydrophilic pendant groups; iii) a polymer that is borderline insoluble in water, i.e., swellable by water but not soluble; iv) introduction of end-group functionality and ring closure to a cyclic form (which the authors call “folding”). For this purpose, they synthesize poly(hydroxyethyl methacrylate) (PHEMA) by ATRP with azide and alkyne end groups with number average DP_n values of 11, 15, 22, (denoted I-PHEMA) and use a high dilution method to obtain the cyclic polymers (denoted f-PHEMA). The dispersities are only modestly narrow ($\bar{D} = 1.35 - 1.39$) and the GPC curves in Figure 1B show a considerable overlap in composition.

The authors then compare what happens with the linear polymer and the folded polymer when methanol solutions are dialyzed against water. The main difference is that the linear polymer precipitates whereas f-PHEMA forms colloiddally stable aggregates. The authors characterized these aggregates by TEM in the dry state on a grid, as well as multiangle dynamic light scattering and X-ray scattering to obtain the hydrodynamic radius R_h and the radius of gyration R_g in solution. These very careful experiments are complemented by coarse-grained molecular dynamics simulations of the conformation and the assembly of a cyclic structure. The submission is accompanied by a number of pretty videos showing interaction of these model structures.

After analysis of this first set of experiments, the authors attach an ATRP initiator to all of the –OH groups of f-PHEMA and grow a series of polystyrene-block-poly(acrylic acid) (PS-b-PAA) pendants from the ring. The assembly of these graft copolymers was compared to that of the corresponding I-PHEMA graft copolymer. Again, the syntheses seem to have been very carefully executed and there is a high level of polymer characterization.

We thank the reviewer for the appreciation of our work and the several pertinent comments and corrections.

Critical comments:

1. The TEM images of the assemblies formed by f-PHEMA in water show elongated structures that are polydisperse in length and variable in width. It would be helpful to the reader if the authors could provide histograms of the length and width distributions. The simulations (Figure 2 E,F) also show a broad dispersity of aggregate sizes and shapes.

Reply: *We thank the reviewer for the suggestion and have added histograms of the length and width distributions in the revised supplementary information (Supplementary Figs. 27-29 E&F). The simulation snapshots in Fig. 2e-f reveal the size variations between approximately 34σ (in the largest aggregate) and about 4σ (in the smallest structures) for $n = 9$, and for $n = 15$ the variation is between 14σ and 3σ . Moreover, because we have used rather mid-sized simulation systems, drawing a size distribution from this system size is a rather nontrivial issue. Note that when we mention the size distribution, we exclude the existence of the isolated single rings in the solution.*

2. It is curious that these structures appear to be characterized by a diffusion coefficient D that is invariant with q^2 (Fig 2C), a behavior expected for spheres. Do the authors have multi-angle static light scattering data for these systems?

Reply: We thank the reviewer for raising this point. Fig. 2c displays the relaxation function due to translational diffusion of the probed moieties over a distance of 336 nm ($=2\pi/q$). The diffusion nature of the motion is proven by the q^2 -dependent relaxation rate, $\Gamma(q)=D(q)q^2$. The observation that $D(q)$ is virtually q -independent (upper inset to Fig. 2c) over the reciprocal space range (0.01 - 0.0224nm^{-1}) or 256-630 nm in the real space implying a magnification, $q^2R_g^2 < 1$. Hence, only a center of mass motion with no discernible contribution of internal structure dynamics is probed by dynamic light scattering. We clarify this point in the revised manuscript.

3. a) The structures formed by the DP_n = 15 polymer was much smaller than those formed by the DP_n = 11 and 22 polymers. I find this surprising and the authors do not comment on the reproducibility of this result.

Reply: The synthesis and self-assembly experiments were repeated several times. Both DLS and TEM data have confirmed that the assemblies formed by the DP_n = 15 polymer was smaller than those formed by the other two polymers. This result indicates that there might be a best suitable size ($n \sim 15$) for f-PHEMA to form a flat and stable ring structure which allows them to pack in a layer-by-layer manner to form wormlike structures. When the ring is larger or smaller, the ring is either not stable or not flat, therefore bigger aggregates were generated.

The simulations are performed at a constant mass density for different n , i.e., with increasing n , the number density ρ of rings decreases. Furthermore, in the initial submitted draft, we had already described that the formation of a structure with a particular size is dictated by competition between the entropic penalty of forming an aggregate from the homogeneous solution and the surface energy reduction. This inherently depends on ρ ; the smaller the ρ , the smaller the sizes of the aggregates. Therefore, the size of an aggregate decreases going from $n = 9$ to $n = 15$, because of decreasing ρ .

For $n = 22$, the aggregate size again increases. This can be attributed to the rather fluffy nature of the ring peripheral stability at $n = 22$. Here, the assemblies form rather mushy objects, without any well-defined structures (with relatively collapsed hydrophobic core). At this point we cannot comment on the critical n value beyond which the well-defined cylindrical aggregates do not exist. This is, however, in our to do list and will be addressed in detail in a future publication. Based on the referee's suggestion we have included additional explanations to discuss this point.

3. b) Some authors (e.g., Chi Wu) use the ratio of R_g/R_h to obtain structural information about objects in solution. Here the authors make no comment about the finding that $R_g/R_h \sim 1$.

Reply: We thank the reviewer for raising this point. It is true that that the value of the ratio R_g/R_h is a hint for the structure, e.g., $R_g/R_h \sim 1.5$ for a coil structure and $(3/5)^{1/2}$ for a spherical globule. In our experiments, we observed $R_g/R_h \sim 1$ (Fig. 2c). Here, however, it is important to mention that the structures seen in the TEM images in Fig. 2b are not single aggregates, rather about 10-20 cylindrical objects stack side-ways to form an aggregate (this

discussion was included in the manuscript). Therefore, in our case R_g/R_h for a bundle reveals the structural features closer to the collapsed objects.

In the simulations, we investigated the formation of the single (isolated) aggregated structures (see Fig. 2e-f). For these isolated structures, we find $R_g/R_h \sim 1.28$ for $n = 9$ and 0.85 for $n = 15$. Based on the referee's suggestion, we have included additional sentences to clarify this point.

3. c) Another feature of the data that the authors do not comment about is the reduction of the integrated intensity in the ^1H NMR spectra of the backbone methyl groups of f-PHEMA15 in mixtures of D2O and MeOD as the D2O content is increased (Figure 2B). This may be a signature of reduced backbone mobility, perhaps as a consequence of the onset of association.

Reply: Thank you for the comment. That could be possible because a reduced mobility broadens signals dramatically (shorter T2 time coming from stronger dipole-dipole interaction) with the result that the integration should be decreased (signals are in the noise due to the broadening or not detectable due to the short life time) or sometimes fully disappeared due to the rigidities. We have added some clarifying sentences in the revised manuscript.

4. The self-assembly results for the series of f-P(HEMA-g-PS-b-PAA) cyclic graft copolymers shown in Figure 3G is very interesting. Significant differences were found as a function of PS and PAA composition, but all are different from that of the linear analogue.

Reply: Thank you for the comment. In Figure 3g, we have shown that cyclic brush polymers with similar compositions demonstrated drastically different self-assembly behaviors compared with their linear analogue. Two important factors are important to achieve this. The first is the cyclic topology and the second is a relatively low weight fraction of PAA (in our work $f_{\text{PAA}} = 7.2\% \sim 16.4\%$). Under these two conditions, cyclic brush polymers form flat and plate-like structures and they cannot fully stabilize themselves in water. Driven by the interactions between the hydrophobic cores, the cyclic brushes stack in a layer-by-layer manner and form cylinders/worms. In other words, the result shown in Figure 3g was realized under well-designed conditions. When the weight fraction of PAA is outside the region, cyclic and linear brush polymers may behave similar (both precipitate when f_{PAA} is too low or solubilize as single brush polymers when f_{PAA} is high enough.)

The self-assembly of cyclic brush copolymers into wormlike aggregates is greatly influenced by the interdigitation of the side chains, which is predominantly due to the PS-based hydrophobic interactions. The larger the concentration of copolymers, the larger the side chain interaction and it also increases the interdigitation, see Supplementary Fig. 65. Therefore, for the higher concentrations, a more network like assembly is observed both in experiments (Fig. 3g-CB4) and in simulations. Based on the referee's suggestions, we have now put forward a detailed explanation in our revised manuscript.

5. All of the structures formed by aggregation in water are complex and hard to describe. This does not diminish the value of these experiments, but one has to be cautious about how one describes this self-assembly process. For example, for f-PHEMA in water, given the relatively broad dispersity, one should probably conclude that the association of these cyclic forms accommodates a broad distribution of contour lengths. Then there are examples where word choice is out of place. For example, in the title, this is a strategy for the emergence of hierarchical nanostructures, but it is not a "general" strategy. The statement on p 2 "folding of polymer chains forming specific secondary structures capable of controlled assembly into anisotropic hierarchical structures" implies a much greater level of control than the authors have achieved. On p 3, the authors write "The physical properties and the

complexity of the propagating structures can be subsequently customized (my italics) by growing a secondary block-copolymer on each HEMA side chain.” It is clear that the secondary block copolymer changes aspects of the self-assembly, but I do not see a design rule that enables the authors to choose a block copolymer composition to reach a predetermined association structure. On the same page, they add “Depending on the composition of the block-copolymer, higher ordered assembly morphology between the unfolded and folded form can be programmed (my italics). Then on page 12 they write “By customizing the block constituents, proportion and length, the architectural outcome can be tuned (my italics). These are unnecessary exaggerations of what they can achieve.

Reply: *We thank the reviewer for the valuable feedback. To better describe the assemblies formed from f-PHEMA, we have added histograms of the length and width distributions in the supplementary information (Supplementary Figs. 27-29 e&f) and revised the manuscript accordingly (... with a distribution of contour lengths in the range of 20~220 nm from...). Additional experiments on the copolymers using Cryo-TEM (Supplementary Fig. S61) were also accomplished to provide further confirmation of the reported morphologies.*

With regards to the title, our initial thoughts were that such a cyclization strategy can be easily extended to other synthetic polymers with different compositions and molecular weights which can diversify the repertoire of hierarchical assemblies with tunable structures. As the referee suggested, the speculation may be too broad and we have therefore removed “as a general strategy” from the title.

For other sentences, we have revised the words used to present our work more precisely. The statement on page 2 “folding of polymer chains forming specific secondary structures capable of controlled assembly into anisotropic hierarchical structures” has been modified to “folding of polymer chains forming unique cyclic structures capable of assembly into wormlike hierarchical structures”. Moreover, “customized” and “programmed” on page 3 have been replaced by “controlled” and “achieved”. The word “tuned” and on page 12 has been replaced by “changed”.

In summary, I like this work. It is an interesting new idea. The execution required careful polymer synthesis, as well as careful and thorough characterization of the assemblies formed. The molecular dynamics simulations were a helpful complement to the experimental work. It represents an important and useful step in addressing an important challenge in the generation of hierarchical structures by synthetic polymers. I recommend acceptance after minor revisions, and I hope that the authors can be more precise in describing what they actually accomplished.

Reviewer #2 (Remarks to the Author):

In this work, Chen et al. presents an interesting approach to prepare (hierarchical) nanostructures via the self-assembly of cyclic polymers/cyclic polymer brushes. The authors started off by showing that a series of short chain cyclic PHEMA polymers (f-PHEMA) can be subjected to a (conventional) dialysis self-assembly process to yield wormlike nanostructures. To highlight the importance of the cyclic polymer topology, the authors showed that linear PHEMA counterparts (l-PHEMA) with similar block lengths cannot be used to access wormlike morphologies. The self-assembly process was scrutinized using various experimental techniques and further elucidated using molecular dynamics simulations. Here, the self-assembly process is presumably driven by phase separation of the hydrophobic methacrylate backbone and the hydrophilic hydroxyl side chains.

In the latter section of the manuscript, the authors expanded on their concept to prepare more structurally complex nanostructures or so-called “hierarchical nanostructures”. This was achieved by, first, converting the cyclic PHEMA polymers (f-PHEMA) into cyclic ATRP macroinitiators, before polymerizing them into cyclic brush polymers (f-P(HEMA-g-PS-b-PAA)). Due to the amphiphilicity of the resulting cyclic brush polymers (f-P(HEMA-g-PS-b-PAA)), they exhibit a different self-assembly behaviour than with cyclic PHEMA polymers (f-PHEMA). Subjection to the same self-assembly procedure (i.e., dialysis from DMF to water) led to the formation of branched wormlike nanostructures. Negative control polymers with different polymer architectures (e.g., linear/linear brushes) were shown to form other non-wormlike morphologies under the same self-assembly conditions. The reported structures here are highly novel, but some further analysis is required to prove their existence.

We thank the reviewer for the appreciation of our work and the several pertinent comments and corrections.

Below are my comments:

Page 7, line 5 – can the authors comment on why the middle ring size gave the smallest R_h values? Looking at Table 1, could this be an effect of the differences in polymer dispersity? f-PHEMA₁₅ has a much higher \bar{D} than the other two folded polymers.

Reply: f-PHEMA₁₅ unexpectedly displays the smallest R_h among the three systems. However, this finding (of small size) cannot be ascribed to the size polydispersity as light scattering inherently emphasizes large moieties. Instead, the low R_h value is consistent with q -independent scattering intensity $I(q)$ which moreover is the lowest among the three samples (Fig. 2c lower inset). Both DLS and TEM data show that the assemblies formed by the $DP_n = 15$ polymer was smaller than those formed by the other two polymers. This result indicates that there might be a best suitable size ($n \sim 15$) for f-PHEMA to form a flat and stable ring structure which allows them to pack in a layer-by-layer manner to form wormlike structures. When the ring is larger or smaller, the ring is either not stable or not flat, therefore bigger aggregates were generated.

This comment of the referee is the same as the comment 3(a) of referee 1. Please see the reply above.

Page 7, line 14 – the authors argue that DLS is not feasible as a characterization technique because the aggregates formed by l-PHEMA are too large. Their TEM data in Figure S27 (~200 nm particles) and Figure S28 (<1 μm), however, suggests otherwise?

Reply: The TEM results do not contradict our statement “For linear PHEMA, DLS experiment

was not possible due to the presence of large aggregates” which is an experimental finding. As explained in the main text, solutions of the linear polymers turned turbid after dialysis for one hour (Supplementary Fig. 26) and some gel-like precipitation was observed three days later. Therefore, the TEM samples of linear PHEMA polymers were prepared by dropping the upper solutions on TEM grids. We have added this note in the revised supplementary information. Dynamic light scattering is very sensitive ($\sim R^6$) to large moieties even at low concentration. We have now added a clarifying sentence in the revised manuscript: “... due to the presence of large aggregates even at low number densities as indicated by the TEM data (Supplementary figs. 27–S29)”.

Page 11, self-assembly process/data – I am a bit confused as to why the self-assembly was carried out in this manner. Typically, when investigating the effect of polymer concentration on polymer self-assembly, one would start with molecularly dissolved polymer solutions at different concentrations and induce self-assembly (e.g., by dialysis against water such as in this work) to generate different polymer morphologies. Here, however, the authors performed self-assembly using a fixed initial polymer concentration of 1.6 mg/mL (according to SI) and added 50 vol% of water (nonsolvent) to the polymer/DMF solution. They then dialyzed this mixture against water and finally adjusted the samples to two concentrations, either to 0.1 mg/mL or 0.4 mg/mL where they observed different apparent morphologies. In some ways, what the authors have done here is they simply started off by generating some form of nanostructure via dialysis at relatively high concentrations (~ 0.8 mg/mL according to the SI) and then subsequently diluting the nanostructures down to 0.1 mg/mL or 0.4 mg/mL. The data presented in Fig. 3 therefore does not adequately highlight the influence of polymer concentration on self-assembly, but rather points towards the fact that the nanostructures formed post-dialysis at ~ 0.8 mg/mL are non-equilibrium structures that dissociate into smaller subunits (apparent as spheres or “nanoobjects” as the authors refer to) at high dilution factors.

Reply: *In the initial submission, we provided the experimental procedure for the preparation of assemblies at 0.4 mg/mL in the supplementary information. For the preparation of solutions at 0.1 mg/mL, we used a similar protocol but a different initial polymer concentration. Therefore, this is indeed the way pointed out by the reviewer. We are sorry for missing the following experimental details:*

For the preparation of solutions at 0.1 mg/mL, 4 mg of polymer was dissolved in 10 mL of DMF at room temperature. The solution was vigorously stirred and 10 mL of deionized water was then added with a speed of 0.2 mL min^{-1} . After stirring for another 2 h, the solution was loaded to into a dialysis membrane (MWCO ~ 1000 Da for block copolymers, MWCO ~ 3500 Da for linear and cyclic brush polymers) and dialyzed against deionized water for three days to completely remove the solvent DMF. The final concentration of polymers was tuned to 0.1 mg mL^{-1} by adding water. We have added these details in the revised supplementary information.

Most importantly, I am currently not sure if the branched wormlike nanostructures are real since the authors have only presented microscopy (TEM/AFM) data at relatively high concentrations where the drying effect in TEM/AFM sample preparation is known to cause particles to “artificially” agglomerate with one another. If possible, the authors should at least present some microscopy data of the particles in their native solution state (e.g., cryo-EM or confocal/fluorescence microscopy; the latter should be easy to conduct since the authors showed in Fig. S33 the possibility of loading a fluorescent dye into their particles). Photographs of the sample solutions would also be very helpful here as highly branched wormlike structures are known to appear as insoluble precipitates in solution that are visible to the naked eye.

Reply: Thank you for these very helpful suggestions. To prove the presence of branched wormlike nanostructures in solution state, we have performed cryo-TEM measurement for **CB-4**. The following images (Supplementary Fig. 61) are consistent with normal TEM and AFM results and they clearly confirm that (branched) wormlike assemblies were formed in solution. We have added these images in the supplementary information.

Supplementary Fig. 61. Cryo-TEM images showing the self-assembly of $f\text{-P}(\text{HEMA-g-PS}_{42}\text{-b-PAA}_5)_{22}$ (**CB-4**) at different concentrations: (A and B) 0.1 mg mL^{-1} ; (C and D) 0.4 mg mL^{-1} .

Furthermore, we tried to stain the sample $f\text{-P}(\text{HEMA-g-PS}_{42}\text{-b-PAA}_5)_{22}$ (**CB-4**, 0.4 mg mL^{-1} in H_2O) for TEM measurement. During the procedure, the sample was first stained with 2% uranyl acetate solution for 45 seconds. After removing the staining solution with a filter paper, the sample was shaken and washed in Milli-Q water for three times (6 seconds for each time). The images in Supplementary Fig. 62 show that the washing/diluting step couldn't destroy the network-like morphology, confirming the good stability of the branched wormlike assemblies.

Supplementary Fig. 62. TEM images showing the morphology of stained samples from $f\text{-P}(\text{HEMA-g-PS}_{42}\text{-b-PAA}_5)_{22}$ (**CB-4**, 0.4 mg mL^{-1} in H_2O).

We also tried TEM measurement under other harsh conditions. The TEM images in Supplementary Fig. 63 show the morphology of $f\text{-P}(\text{HEMA-g-PS}_{42}\text{-b-PAA}_5)_{22}$ (**CB-4**, 0.4 mg mL^{-1} in H_2O) after sonication for one hour. Although we can see that these assemblies become shorter which is quite understandable, they remained the branched wormlike shape. The shortening of these structures after sonication also confirms from a different perspective that the branched wormlike assemblies in Figure 3g were not generated during the sample preparation process. Otherwise, the lengths of the structures after sonication should not change. In addition, we also found that these assemblies were stable even after storage at room temperature for two months (Supplementary Fig. 64).

Supplementary Fig. 63. TEM images of the self-assembled solution based on $f\text{-P}(\text{HEMA-g-PS}_{42}\text{-b-PAA}_5)_{22}$ (**CB-4**, 0.4 mg mL^{-1} in H_2O) after sonication for one hour.

Supplementary Fig. 64. TEM images of the self-assembled solution based on *f*-P(HEMA-*g*-PS₄₂-*b*-PAA₅)₂₂ (**CB-4**, 0.4 mg mL⁻¹ in H₂O) after storage at room temperature for two months.

We have also attempted confocal laser scanning microscopy. However, we were not able to resolve the structures as they were too small. Regarding the sample solutions, we could observe a small amount of suspended matter in solution when the assembly concentration is 0.4 mg mL⁻¹. However, the suspended matter is different from the precipitates generated from linear brush polymers as shown in Supplementary Fig. 51. It is nearly colorless and can hardly be captured by a camera. Collectively, we can conclude that the branched wormlike nanostructures in Fig. 3g were already formed in solution and they demonstrated good stability.

Also, I am curious as to why no DLS data was provided for the nanostructures in Fig. 3. I find this very surprising since DLS was used to thoroughly characterize the wormlike nanostructures formed by *f*-PHEMA (see Fig. 2C and Fig. S32).

Reply: Thank you for the comment. These cyclic brushes have a strong trend to form large aggregates even in a good solvent. For example, the following figure shows the DLS result of **CB-3** dissolved in DMF at a low concentration of 0.1 mg mL⁻¹. Two portions were detected ($R_{h1} = 4.5$ nm, $R_{h2} = 28$ nm). The smaller one can be ascribed to single cyclic brushes and the bigger one is from aggregates.

For nanostructures formed in water after dialysis, the measurement was not possible. As mentioned above, dynamic light scattering is very sensitive ($\sim R^6$) to large moieties even at low concentration. This is also a strong indication that the hierarchical structures observed from TEM and AFM images are already formed in solution.

Regarding the self-assembly of the cyclic polymer brushes – Can the authors comment on the formation mechanism behind the branched wormlike nanostructures? I struggle to understand how individual polymer subunits (i.e., nanoobjects in Fig. 3F/G) would preferentially stack in an anisotropic fashion form cylinders/worms or even branch out into fractal structures. These polymer subunits are supposedly decorated with a PAA brush corona. Wouldn't the particles prefer to repel one another due to electrostatic repulsion in addition to the brush conformation?

Reply: *Thank you for the comment. For the self-assembly of the cyclic polymer brushes, PAA acts as the hydrophilic block and PS as the hydrophobic block. At neutral conditions, PAA chains do not repel each other due to limited ionization, which is the same situation in those classical studies about the self-assembly of PS-b-PAA block copolymers (Chem Soc Rev, 2012, 41, 5969; Science, 1995, 268, 1728; JACS, 1996, 118, 3168). In our system, the weight fraction of PAA is relatively low ($f_{\text{PAA}} = 7.2\% \sim 16.4\%$). In this case, PAA cannot fully stabilize the brush polymers in water. Because of their folded cyclic topology, they form flat and plate-like structures first. Driven by the hydrophobic interactions of the cores, the cyclic brushes stack in a layer-by-layer manner to form cylinders/worms.*

*The self-assembly of cyclic brush copolymers into wormlike aggregates is greatly influenced by the interdigitation of the side chains, which is predominantly due to the PS-PS interaction between the neighboring clusters. The larger the concentration of copolymers, the larger the side chain interaction and it also increases the interdigitation, see Supplementary Fig. 65. Therefore, for the higher concentrations, a more network like assembly is observed both in experiments (Fig. 3g-**CB4**) and in simulations. We have added the discussion about the formation mechanism in the revised manuscript.*

At the present stage, I find the first half of the manuscript to be well-written and substantiated by convincing data. However, the second half of the manuscript (in particular, the section on the self-assembly cyclic polymer brushes) needs more data. I strongly suggest the authors perform (i) additional characterization on their so-called “wormlike assemblies” and “hierarchical structures” to refute the possibility that these morphologies are imaging artifacts, and (ii) elaborate on their formation mechanism, if they are indeed real.

Editorial Note: Reviewer #1 was unable to re-review the paper so Reviewer #2 was asked to look over all of the authors responses to all queries by both reviewers

REVIEWERS' COMMENTS

Reviewer #2 (Remarks to the Author):

The authors provided an excellent revision. I went through all the comments and they have addressed everything to my satisfaction. In particular the cryoTEM is a strong addition.

I have two very minor comments that can be easily addressed:

The authors state that they repeated the experiments several times to confirm that DP_n=15 forms indeed smaller aggregates. It would have been useful to have the standard deviation included as this would highlight the robustness of the approach.

The authors answered well the question why they did not include the DLS data for the nanostructures in Fig. 3. It would have been however useful to add a sentence to the main text as to highlight the low solubility in DMF, which led to large aggregates in water. This is an interesting information for the reader.

Point by Point Response

Reviewer #2 (Remarks to the Author):

The authors provided an excellent revision. I went through all the comments and they have addressed everything to my satisfaction. In particular the cryoTEM is a strong addition.

I have two very minor comments that can be easily addressed:

The authors state that they repeated the experiments several times to confirm that $DP_n=15$ forms indeed smaller aggregates. It would have been useful to have the standard deviation included as this would highlight the robustness of the approach.

Reply: *Thank you for your kind comments. Indeed, the standard deviation of each of the polymers at $DP_n = 11, 15, 22$ in light scattering has been reflected the manuscript to show that the polymers produced at $DP_n = 15$ reproducibly forms smaller aggregates.*

The authors answered well the question why they did not include the DLS data for the nanostructures in Fig. 3. It would have been however useful to add a sentence to the main text as to highlight the low solubility in DMF, which led to large aggregates in water. This is an interesting information for the reader.

Reply: *Thank you for the suggestion. We have added the following sentence in the main text (page 9 last paragraph): "Although the nature of the block-copolymer (LB) and its linear polymerized form (BC) exhibit low solubility already in DMF featuring large aggregates upon addition of water, the cyclization is able to alleviate these effects and create stable assemblies."*